# PROCEEDINGS A

statistics

independence, Pearson's $\chi^2$-test, *G*-test, permutation test, statistic, Fisher's exact test

**Author for correspondence:**
Richard J. Samworth
e-mail: r.samworth@statslab.cam.ac.uk

# USP: an independence test that improves on Pearson's chi-squared and the *G*-test

Thomas B. Berrett[1] and Richard J. Samworth[2]

[1]University of Warwick, Coventry CV4 7AL, UK
[2]University of Cambridge, Cambridge CB2 1TN, UK

RJS, 0000-0003-2426-4679

We present the *U*-statistic permutation (USP) test of independence in the context of discrete data displayed in a contingency table. Either Pearson's $\chi^2$-test of independence, or the *G*-test, are typically used for this task, but we argue that these tests have serious deficiencies, both in terms of their inability to control the size of the test, and their power properties. By contrast, the USP test is guaranteed to control the size of the test at the nominal level for all sample sizes, has no issues with small (or zero) cell counts, and is able to detect distributions that violate independence in only a minimal way. The test statistic is derived from a *U*-statistic estimator of a natural population measure of dependence, and we prove that this is the unique minimum variance unbiased estimator of this population quantity. The practical utility of the USP test is demonstrated on both simulated data, where its power can be dramatically greater than those of Pearson's test, the *G*-test and Fisher's exact test, and on real data. The USP test is implemented in the R package USP.

## 1. Introduction

Pearson's $\chi^2$-test of independence [1] is one of the most commonly used of all statistical procedures. It is typically employed in situations where we have discrete data consisting of independent copies of a pair $(X, Y)$, with $X$ taking the value $x_i$ with probability $q_i$, for $i = 1, \ldots, I$, and $Y$ taking the value $y_j$ with probability $r_j$, for $j = 1, \ldots, J$. For example, $X$ might represent marital status, taking values 'Never married', 'Married', 'Divorced', 'Widowed' and $Y$ might represent level of education,

**Table 1.** Contingency table summarizing the marital status and education level of 300 survey respondents. Source: https://www.spss-tutorials.com/chi-square-independence-test/.

| | Middle school or Lower | High school | Bachelor's | Master's | PhD or Higher |
|---|---|---|---|---|---|
| Never married | 18 | 36 | 21 | 9 | 6 |
| Married | 12 | 36 | 45 | 36 | 21 |
| Divorced | 6 | 9 | 9 | 3 | 3 |
| Widowed | 3 | 9 | 9 | 6 | 3 |

with values 'Middle school or lower', 'High school', 'Bachelor's', 'Master's', 'PhD or higher', so that $I = 4$ and $J = 5$. From a random sample of size $n$, we can summarize the resulting data $(X_1, Y_1), \ldots, (X_n, Y_n)$ in a contingency table with $I$ rows and $J$ columns, where the $(i, j)$th entry $o_{ij}$ of the table denotes the observed number of data pairs equal to $(x_i, y_j)$; see table 1 for an illustration.

Writing $p_{ij} = P(X = x_i, Y = y_j)$ for the probability that an observation falls in the $(i, j)$th cell, a test of the null hypothesis $H_0$ that $X$ and $Y$ are independent is equivalent to testing whether $p_{ij} = q_i r_j$ for all $i, j$. Letting $o_{i+}$ denote the number of observations falling in the $i$th row and $o_{+j}$ denote the number in the $j$th column, Pearson's famous formula can be expressed as

$$\chi^2 = \sum_{i=1}^{I} \sum_{j=1}^{J} \frac{(o_{ij} - e_{ij})^2}{e_{ij}}, \tag{1.1}$$

where $e_{ij} = o_{i+} o_{+j} / n$ is the 'expected' number of observations in the $(i, j)$th cell under the null hypothesis. Usually, for a test of size approximately $\alpha$, the $\chi^2$ statistic is compared with the $(1 - \alpha)$-level quantile of the $\chi^2$ distribution with $(I - 1)(J - 1)$ degrees of freedom[1]. For instance, for the data in table 1, we find that $\chi^2 = 23.6$, corresponding to a $p$-value of 0.0235. This analysis would therefore lead us to reject the null hypothesis at the 5% significance level, but not at the 1% level.

Pearson's $\chi^2$-test is so well established that we suspect many researchers would rarely pause to question whether or not it is a good test. The formula (1.1) arises as a second-order Taylor approximation to the generalized likelihood ratio test, or $G$-test as it is now becoming known (e.g. [4]):

$$G = 2 \sum_{i=1}^{I} \sum_{j=1}^{J} o_{ij} \log \frac{o_{ij}}{e_{ij}}.$$

The $G$-test statistic is compared with the same $\chi^2$ quantile as Pearson's statistic, and its use is advocated in certain application areas, such as computational linguistics [5]. There is also a second motivation for the statistic (1.1), which relies on the idea of the $\chi^2$ *divergence* between two probability distributions $P = (p_{ij})$ and $P' = (p'_{ij})$ for our pair $(X, Y)$:

$$\chi^2(P, P') = \sum_{i=1}^{I} \sum_{j=1}^{J} \frac{(p_{ij} - p'_{ij})^2}{p'_{ij}}. \tag{1.2}$$

The word 'divergence' here is used by statisticians to indicate that $\chi^2(P, P')$ is a quantity that behaves in some ways like a (squared) distance, e.g. $\chi^2(P, P')$ is non-negative, and is zero if and only if $P = P'$, but does not satisfy all of the properties that we would like a genuine notion of distance to have. For instance, it is not symmetric in $P$ and $P'$—we can have $\chi^2(P, P') \neq \chi^2(P', P)$. Pearson's statistic can be regarded as the natural empirical estimate of the $\chi^2$ divergence between the joint distribution $P = (p_{ij})$ and the product $P'$ of the marginal distributions $(q_i)$ and $(r_j)$. This

---

[1]As an interesting historical footnote, Pearson's original calculation of the number of degrees of freedom contained an error, which was corrected by Fisher [2]; e.g. Lehmann & Romano [3, p. 741].

**Table 2.** Expected frequencies for the data in table 1, with the $(i, j)$th entry computed as $e_{ij} = o_{i+}o_{+j}/n$.

|  | Middle school or Lower | High school | Bachelor's | Master's | PhD or Higher |
|---|---|---|---|---|---|
| Never married | 11.7 | 27 | 25.2 | 16.2 | 9.9 |
| Married | 19.5 | 45 | 42 | 27 | 16.5 |
| Divorced | 3.9 | 9 | 8.4 | 5.4 | 3.3 |
| Widowed | 3.9 | 9 | 8.4 | 5.4 | 3.3 |

makes some sense when we recall that the null hypothesis of independence holds if and only if the joint distribution is equal to the product of the marginal distributions (e.g. [6, theorem 3B]).

Nevertheless, both Pearson's $\chi^2$-test and the $G$-test suffer from three major drawbacks:

1. The tests do not in general control the probability of Type I error at the claimed level. In fact, we show in appendix Aa that even in the simplest setting of a $2 \times 2$ table, and no matter how large the sample size $n$, it is possible to construct a joint distribution that satisfies the null hypothesis of independence, but for which the probability of Type I error is far from the desired level! Practitioners are aware of this deficiency of Pearson's test and the $G$-test (e.g. [7, p. 40]), but our example provides an explicit demonstration.
2. If there are no observations in any row or column of the table, then both test statistics are undefined.
3. Perhaps most importantly, the power properties of both Pearson's $\chi^2$-test and the $G$-test are poorly understood. The well-known optimality of likelihood ratio tests in many settings where the null hypothesis consists of a single distribution, which follows from the famous Neyman–Pearson lemma [8], does not translate over to independence tests, where the null hypothesis is *composite*—i.e. there is more than one distribution that satisfies its constraints.

The first two concerns mentioned above are related to small cell counts, which are known to cause issues for both Pearson's $\chi^2$-test and the $G$-test. Indeed, elementary Statistics textbooks typically make sensible but *ad hoc* recommendations, such as:

> [Pearson's $\chi^2$-test statistic] approximately follows the $\chi^2$ distribution … provided that (1) all expected frequencies are greater than or equal to 1 and (2) no more than 20% of the expected frequencies are less than 5 (Sullivan III [9, p. 623]).
>
> The $X^2$ statistic has approximately a $\chi^2$ distribution, for large $n$ … The $\chi^2$ approximation improves as $\{\mu_{ij}\}$ increase, and $\{\mu_{ij} \geq 5\}^2$ is usually sufficient for a decent approximation (Agresti [7, p. 35]).

Unfortunately, these recommendations (and others in different sources) may be contradictory, leaving the practitioner unsure of whether or not they can apply the tests. For instance, for the data in table 1, we obtain the expected frequencies given in table 2. From this table, we see that all of the expected frequencies are greater than 1 but four of the 20 cells, i.e. exactly 20%, have expected frequencies less than 5, meaning that this table just satisfies Sullivan, III's criteria, but it does not satisfy Agresti's.

Fortunately, there is a well-known, though surprisingly rarely applied, fix for the first numbered problem above, for both Pearson's test and the $G$-test: we can obtain the critical value via a permutation test. We will discuss permutation tests in detail in §2, but for now it suffices to note that this approach guarantees that the tests control the size of the tests at the nominal level $\alpha$, in the sense that for every sample size $n$, the tests have Type I error probability no greater than $\alpha$.

---

$^2\{e_{ij} \geq 5\}$ in our notation.

Our second concern above would typically be handled by removing rows or columns with no observations. If such a row or column had positive probability, however, then this amounts to changing the test being conducted. For instance, if we suppose for simplicity that the $I$th row has no observations, but $q_I > 0$, then we are only testing the null hypothesis that $p_{ij} = q_i r_j$ for $i = 1, \ldots, I - 1$ and $j = 1, \ldots, J$. This is not sufficient to verify that $X$ and $Y$ are independent.

It is, however, the third drawback listed above that is arguably the most significant. When the null hypothesis is false, we would like to reject it with as large a probability as possible. It is too much to hope here that a single test of a given size will have the greatest power to reject every departure from the null hypothesis. If we have two reasonable tests, $A$ and $B$, then typically Test $A$ will be better at detecting departures from the null hypothesis of a particular form, while Test $B$ will have greater power for other alternatives. Even so, it remains important to provide guarantees on the power of a proposed test to justify its use in practice, as we discuss in §2, yet the seminal monograph on statistical tests of Lehmann & Romano [3] is silent on the power of both Pearson's test and the $G$-test.

The aim of this work, then, is to describe an alternative test of independence, called the USP test (short for $U$-Statistic Permutation test), which simultaneously remedies all of the drawbacks mentioned above. Since it is a permutation test, it controls the Type I error probability at the desired level[3] for every sample size $n$. It has no problems in handling small (or zero) cell counts. Finally, we present its strong theoretical guarantees, which come in two forms: first, the USP test is able to detect departures that are minimally separated, in terms of the sample size-dependent rate, from the null hypothesis. Second, we show that the USP test statistic is derived from the unique minimum variance unbiased estimator of a natural measure of dependence in a contingency table. To complement these theoretical results, we present several numerical comparisons between the USP test and both Pearson's test and the $G$-test, as well as another alternative, namely Fisher's exact test (e.g. [7, §2.6]), which provide further insight into the departures from the null hypothesis for which the USP test will represent an especially large improvement.

The USP test was originally proposed by Berrett *et al.* [10], who worked in a much more abstract framework that allows categorical, continuous and even functional data to be treated in a unified manner. Here, we focus on the most important case for applied science, namely categorical data, and seek to make the presentation as accessible as possible, in the hope that it will convince practitioners of the merits of the approach.

## 2. The USP test of independence

One starting point to motivate the USP test is to note that many of the difficulties of Pearson's $\chi^2$-test and the $G$-test stem from the presence of the $e_{ij}$ terms in the denominators of the summands. When $e_{ij}$ is small, this can make the test statistics rather unstable to small perturbations of the observed table. This suggests that a more natural (squared) distance measure than the $\chi^2$-divergence (1.2) is

$$D(P, P') = \sum_{i=1}^{I} \sum_{j=1}^{J} (p_{ij} - p'_{ij})^2.$$

Unlike the $\chi^2$-divergence, this definition is symmetric in $P$ and $P'$. In independence testing, we are interested in the case where $P'$ is the product of the marginal distributions of $X$ and $Y$, i.e. $p'_{ij} = q_i r_j$. We can therefore define a measure of dependence in our contingency table by

$$D = \sum_{i=1}^{I} \sum_{j=1}^{J} (p_{ij} - q_i r_j)^2.$$

---

[3]In fact, this represents an important advantage of permutation tests over the bootstrap (another natural choice to obtain $p$-values) in independence-testing problems.

Under the null hypothesis of independence, we have $p_{ij} = q_i r_j$ for all $i, j$, so $D = 0$. In fact, the only way we can have $D = 0$ is if $X$ and $Y$ are independent. More generally, the non-negative quantity $D$ represents the extent of the departure of $P$ from the null hypothesis of independence.

Note that $p_{ij}$, $q_i$ and $r_j$ are population-level quantities, so we cannot compute $D$ directly from our observed contingency table. We can, however, seek to estimate it, and indeed this is the approach taken by Berrett *et al.* [10]. To understand the main idea, suppose for simplicity that $X$ can take values from 1 to $I$, and $Y$ and take values from 1 to $J$. Consider the function

$$h\big((x_1, y_1), (x_2, y_2), (x_3, y_3), (x_4, y_4)\big)$$

$$= \sum_{i=1}^{I} \sum_{j=1}^{J} \big(1_{\{x_1=i, y_1=j\}} 1_{\{x_2=i, y_2=j\}} - 2 \cdot 1_{\{x_1=i, y_1=j\}} 1_{\{x_2=i\}} 1_{\{y_3=j\}}$$

$$+ 1_{\{x_1=i\}} 1_{\{y_2=j\}} 1_{\{x_3=i\}} 1_{\{y_4=j\}}\big),$$

where, for instance, the indicator function $1_{\{x_1=i, y_1=j\}}$ is 1 if $x_1 = i$ and $y_1 = j$, and is zero otherwise. We claim that $h((X_1, Y_1), (X_2, Y_2), (X_3, Y_3), (X_4, Y_4))$ is an unbiased estimator of $D$; this follows because

$$Eh\big((X_1, Y_1), (X_2, Y_2), (X_3, Y_3), (X_4, Y_4)\big)$$

$$= \sum_{i=1}^{I} \sum_{j=1}^{J} \{P(X_1 = i, Y_1 = j)P(X_2 = i, Y_2 = j) - 2P(X_1 = i, Y_1 = j)P(X_2 = i)P(Y_3 = j)$$

$$+ P(X_1 = i)P(Y_2 = j)P(X_3 = i)P(Y_4 = j)\}$$

$$= \sum_{i=1}^{I} \sum_{j=1}^{J} (p_{ij}^2 - 2p_{ij}q_i r_j + q_i^2 r_j^2) = \sum_{i=1}^{I} \sum_{j=1}^{J} (p_{ij} - q_i r_j)^2 = D.$$

However, $h((X_1, Y_1), (X_2, Y_2), (X_3, Y_3), (X_4, Y_4))$ on its own is not a good estimator of $D$, because it only uses the first four data pairs, so it would have high variance. Instead, what we can do is to construct an estimator $\widehat{D}$ of $D$ as the average value of $h$ as the indices of its arguments range over all possible sets of four distinct data pairs within our dataset. In other words,

$$\widehat{D} = \frac{1}{4!\binom{n}{4}} \sum_{(i_1, i_2, i_3, i_4)} h\big((X_{i_1}, Y_{i_1}), (X_{i_2}, Y_{i_2}), (X_{i_3}, Y_{i_3}), (X_{i_4}, Y_{i_4})\big),$$

where the sum is over all distinct indices $i_1, i_2, i_3, i_4$ between 1 and $n$. Thus, we have $n$ choices for the first data pair, $n - 1$ choices for the second data pair, $n - 2$ for the third and $n - 3$ for the fourth, meaning that $\widehat{D}$ is an average of $n(n-1)(n-2)(n-3) = 4!\binom{n}{4}$ terms, each of which has the same distribution, and therefore in particular, the same expectation, namely $D$. It follows that $\widehat{D}$ is an unbiased estimator of $D$, but since it is an average, it will have much smaller variance than the naive estimator $h((X_1, Y_1), (X_2, Y_2), (X_3, Y_3), (X_4, Y_4))$.

Estimators constructed as averages of so-called *kernels h* over all possible sets of distinct data points are called *U-statistics*, and the fact that there are four data pairs to choose means that $\widehat{D}$ is a *fourth-order U*-statistic. For more information about *U*-statistics, see, for example, Serfling [11, ch. 5].

The final formula for $\widehat{D}$ does simplify somewhat, but remains rather unwieldy; it is given for the interested reader in appendix Ab. Fortunately, and as we explain in detail below, for the purposes of constructing a permutation test of independence, only part of the estimator is relevant. This leads to the definition of the USP test statistic, for $n \geq 4$, as

$$\widehat{U} = \frac{1}{n(n-3)} \sum_{i=1}^{I} \sum_{j=1}^{J} (o_{ij} - e_{ij})^2 - \frac{4}{n(n-2)(n-3)} \sum_{i=1}^{I} \sum_{j=1}^{J} o_{ij} e_{ij}. \tag{2.1}$$

This formula appears a little complicated at first glance, so let us try to understand how the terms arise. Notice that $o_{ij}/n$ is an unbiased estimator of $p_{ij}$, and, under the null hypothesis, $e_{ij}/n$ is an unbiased estimator of $q_i r_j$. Thus the first term in (2.1) can be regarded as the leading order term in the estimate of $D$. The second term (2.1) can be seen as a higher-order bias correction term that accounts for the fact that the same data are used to estimate $p_{ij}$ and $q_i r_j$; in other words, $o_{ij}/n$ and $e_{ij}/n$ are dependent.

To carry out the USP test, we first compute the statistic $\widehat{U} = \widehat{U}(T)$ on the original data $T = \{(X_1, Y_1), \ldots, (X_n, Y_n)\}$. We then choose $B$ to be a large integer ($B = 999$ is a common choice), and, for each $b = 1, \ldots, B$, generate an independent permutation $\sigma^{(b)}$ of $\{1, \ldots, n\}$ uniformly at random among all $n!$ possible choices. This allows us to construct permuted datasets[4] $T^{(b)} = \{(X_1, Y_{\sigma^{(b)}(1)}), \ldots, (X_n, Y_{\sigma^{(b)}(n)})\}$, and to compute the test statistics $\widehat{U}^{(b)} = \widehat{U}(T^{(b)})$ that we would have obtained if our data were $T^{(b)}$ instead of $T$. The key point here is that, since the original data consisted of $n$ independent pairs, we certainly know for instance that $X_1$ and $Y_{\sigma^{(b)}(1)}$ are independent under the null hypothesis. Thus the pseudo-test statistics $\widehat{U}^{(1)}, \ldots, \widehat{U}^{(B)}$ can be regarded as being drawn from the null distribution of $\widehat{U}$. This means that, in order to assess whether or not our real test statistic $\widehat{U}$ is extreme by comparison with what we would expect under the null hypothesis, we can compute its rank among all $B + 1$ test statistics $\widehat{U}, \widehat{U}^{(1)}, \ldots, \widehat{U}^{(B)}$, where we break ties at random. If we seek a test of Type I error probability $\alpha$, then we should reject the null hypothesis of independence if $\widehat{U}$ is at least the $\alpha(B + 1)$th largest of these $B + 1$ test statistics.

It is a standard fact (e.g. [12, lemma 2]) about permutation tests such as this that, even when the null hypothesis is composite (as is the case for independence tests in contingency tables), the Type I error probability of the test is at most $\alpha$, for all sample sizes for which the test is defined ($n \geq 4$ in our case). Comparing (2.1) with the long formula for $\widehat{D}$ in (A 2), we see that we have ignored some additional terms that only depend on the observed row and column totals $o_{i+}$ and $o_{+j}$. To understand why we can do this, imagine that instead of computing $\widehat{U}, \widehat{U}^{(1)}, \ldots, \widehat{U}^{(B)}$, we instead computed the corresponding quantities $\widehat{D}, \widehat{D}^{(1)}, \ldots, \widehat{D}^{(B)}$, on the original and permuted datasets, respectively. Since the row and column totals $o_{i+}$ and $o_{+j}$ are identical for the permuted datasets[5] as for the original data, we see that the rank of $\widehat{U}$ among $\widehat{U}, \widehat{U}^{(1)}, \ldots, \widehat{U}^{(B)}$ is the same as the rank of $\widehat{D}$ among $\widehat{D}, \widehat{D}^{(1)}, \ldots, \widehat{D}^{(B)}$. Therefore, when working with the simplified test statistic $\widehat{U}$, we will reject the null hypothesis if and only if we would also reject the null hypothesis when working with the full unbiased estimator $\widehat{D}$.

As mentioned in the introduction, Berrett *et al.* [10] showed that the USP test is able to detect alternatives that are minimally separated from the null hypothesis, as measured by $D$. More precisely, given an arbitrarily small $\epsilon > 0$, we can find $C > 0$, depending only on $\epsilon$, such that for any joint distribution $P$ with $D \geq Cn^{-1}$, the sum of the two error probabilities of the USP test is smaller than $\epsilon$. Moreover, no other test can do better than this in terms of the rate: again, given any $\epsilon > 0$ and any other test, there exists $c > 0$, depending only on $\epsilon$, and a joint distribution $P$ with $D \geq cn^{-1}$, such that the sum of the two error probabilities of this other test is greater than $1 - \epsilon$. This result provides a sense in which the USP test is optimal for independence testing for categorical data.

To complement the result above, we now derive a new and highly desirable property of the $U$-statistic $\widehat{D}$ in (A 2).

**Theorem 2.1.** *The statistic $\widehat{D}$ is the unique minimum variance unbiased estimator of $D$.*

The proof of theorem 2.1 is given in appendix Ac. Once one accepts that $D$ is a sensible measure of dependence in our contingency table, theorem 2.1 is reassuring in that it provides a sense in which $\widehat{D}$ is a very good estimator of $D$. Since $\widehat{U}$ is equally as good a test statistic as $\widehat{D}$, as explained above, this provides further theoretical support for the USP test.

[4]In fact, as shown in Berrett *et al.* [10], we only require the original cell counts $o_{ij}$ to compute the cell counts $o_{ij}^{(b)}$ for the permuted data. This dramatically simplifies the computation of the contingency tables for permuted datasets.

[5]Interestingly, this is another advantage of using permutations as opposed to the bootstrap to generate *p*-values.

# 3. Numerical results

## (a) Software

The USP test is implemented in the R package USP [13]. Once the package has been installed and loaded, it can be run on the data in table 1 as follows:

```
> Data = matrix(c(18,12,6,3,36,36,9,9,21,45,9,9,9,36,3,6,6,21,3,3),4,5)
> USP.test(Data)
```

As with all permutation tests, the $p$-values obtained using the USP test will typically not be identical on different runs with the same data, due to the randomness of the permutations. The default choice of $B$ for the USP.test function is 999, which in our experience, yields quite stable $p$-values over different runs. This stability could be increased by running

```
> USP.test(Data,B = 9999)
```

for example (although this will increase the computational time). Using $B = 999$ yielded a $p$-value of 0.001, so with the USP test, we would reject the null hypothesis of independence even at the 1% level. For comparison, the $G$-test $p$-value is 0.0205, while Fisher's exact test has a $p$-value of 0.02, so like Pearson's test, they fail to reject the null hypothesis at the 1% level.

## (b) Simulated data

In this subsection, we compare the performance of the USP test, Pearson's test, the $G$-test and Fisher's exact test on various simulated examples. For each example, we need to choose the sample size $n$, as well as the number of rows $I$ and columns $J$ of our contingency table. However, the most important choice is that of the type of alternative that we seek to detect. Recall that the null hypothesis holds if and only if $p_{ij} = q_i r_j$ for all $i, j$. There are many ways in which this family of equalities might be violated, but it is natural to draw a distinction between situations where only a small number of the equalities fail to hold (*sparse* alternatives), and those where many fail to hold (*dense* alternatives). It turns out that the smallest possible non-zero number of violations is four, and our initial example will consider such a setting.

The starting point for this first example is a family of cell probabilities that satisfy the null hypothesis

$$p_{ij} = \frac{2^{-(i+j)}}{(1 - 2^{-I})(1 - 2^{-J})}, \tag{3.1}$$

for $i = 1, \ldots, I$ and $j = 1, \ldots, J$. A pictorial representation of these cell probabilities is given in figure 1, which illustrates that the cell probability halves every time we move one cell to the right, or one cell down in the table. The corresponding marginal probabilities for the $i$th row and $j$th column are $q_i = 2^{-i}/(1 - 2^{-I})$ and $r_j = 2^{-j}/(1 - 2^{-J})$, respectively. Now, to construct a family of cell probabilities that can violate the null hypothesis in a small number of cells, we will fix $\epsilon \geq 0$ and define modified cell probabilities

$$p_{ij}^{(\epsilon)} = \begin{cases} p_{ij} + \epsilon & \text{if } (i,j) = (1,1) \text{ or } (i,j) = (2,2) \\ p_{ij} - \epsilon & \text{if } (i,j) = (1,2) \text{ or } (i,j) = (2,1) \\ p_{ij} & \text{otherwise.} \end{cases}$$

Note that $p_{ij}^{(0)}$ is just the original cell probability $p_{ij}$, and that, for $\epsilon > 0$, we can consider the new cell probabilities to be a sparse perturbation of the original ones, because we only change the probabilities in the top-left block of four cells. The parameter $\epsilon$, which needs to be chosen small enough that all of the cell probabilities lie between 0 and 1, controls the extent of the dependence

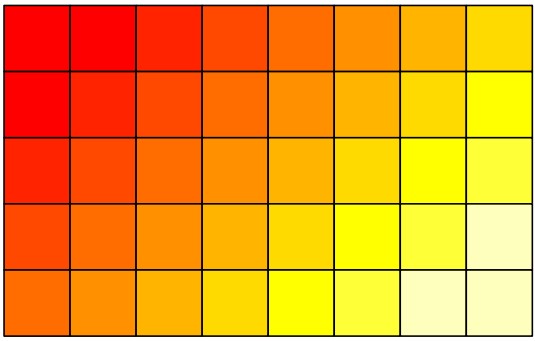

**Figure 1.** Pictorial representation of the cell probabilities in (3.1). (Online version in colour.)

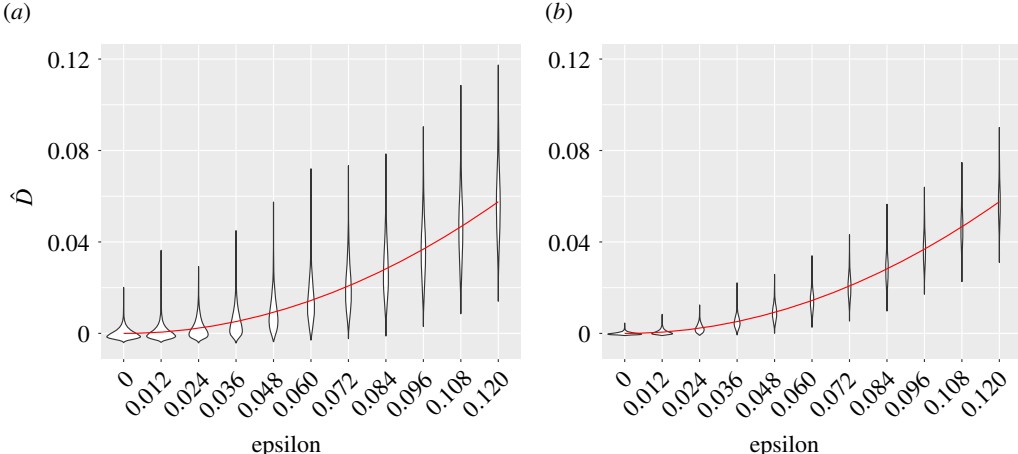

**Figure 2.** Violin plots of the values of $\widehat{D}$ with $I = 5$, $J = 8$ and with $n = 100$ (a) and $n = 400$ (b) for different values of $\epsilon$. The function $f(\epsilon) = 4\epsilon^2$ is shown as a red line. (Online version in colour.)

in the table; in fact, we can calculate that our dependence measure $D$ is equal to $4\epsilon^2$ in this example.

We first study how well our estimator $\widehat{D}$ is able to estimate $D$. In figure 2, we present violin plots giving a graphical representation of the values of $\widehat{D}$ obtained from 10 000 contingency tables generated with $I = 5$ and $J = 8$, for 11 different values of $\epsilon$ and for $n = 100$ and $n = 400$; we also plot the quadratic function $f(\epsilon) = 4\epsilon^2$. This figure provides numerical support for the fact that $\widehat{D}$ is an unbiased estimator of $D$, and illustrates the way that the variance of $\widehat{D}$ decreases as the sample size increases from 100 to 400.

Next, we turn to the size and power of the USP test, and compare them with those of Pearson's test, the $G$-test and Fisher's exact test. Figure 3 shows the way in which the power of these tests increases with $\epsilon$, for a test of nominal size 5%, with $n = 100$ (the corresponding plot with $n = 400$, which is qualitatively similar, is given in figure 8). For both Pearson's test and the $G$-test, we plot power curves for both the version of the test that takes the critical value from the $\chi^2$ distribution with $(I - 1)(J - 1)$ degrees of freedom, and the version that obtains the critical value using a permutation test, like the USP test. Here and below, for all permutation tests, we took $B = 999$.

The most striking feature of figure 3 is the extent of the improvement of the USP test over its competitors. When $\epsilon = 0.06$, for instance, the USP test is able to reject the null hypothesis in 89% of the experiments, whereas even the better (permutation) version of Pearson's test only achieves

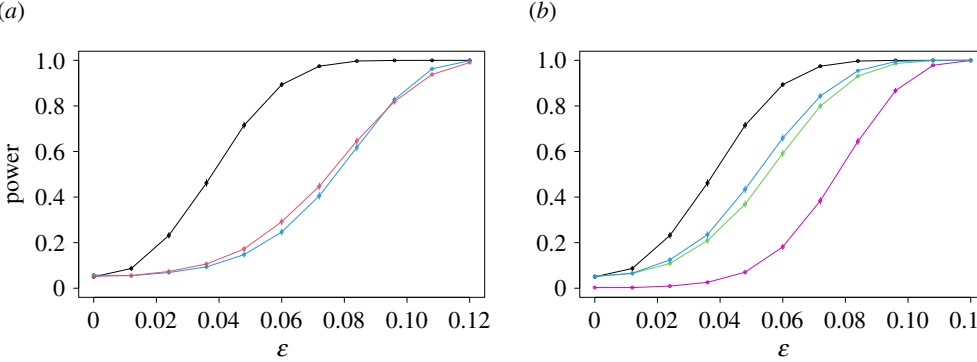

**Figure 3.** Power curves of the USP test in the sparse example, compared with Pearson's test (*a*) and both the *G*-test and Fisher's exact test (*b*). In each case, the power of the USP test is given in black. The power functions of the $\chi^2$ quantile versions of the first two comparators are shown in blue (*a*) and purple (*b*), while those of the permutation versions of these tests are given in red (*a*) and green (*b*). The power curve of Fisher's exact test is shown in cyan on the right. In this plot, as in the other power curve plots, vertical lines through each data point indicate three standard errors (though with 10 000 repetitions, these are barely visible). (Online version in colour.)

a power of 29%. The permutation version of the *G*-test and Fisher's exact test do slightly better in this example, achieving powers of 59% and 66% respectively, but remain uncompetitive with the USP test. The version of the *G*-test that uses the chi-squared quantile for the critical value performs poorly in this example, because it is conservative (i.e. its true size is less than the nominal level 5% level). This can be seen from the fact that the leftmost data point of the purple curve on the right-hand plot in figure 3, which corresponds to the proportion of the experiments for which the null hypothesis was rejected when it was true, is considerably less than 5%. It is also straightforward to construct examples for which the versions of Pearson's test and the *G*-test that use the $\chi^2$ quantile are anti-conservative (i.e. do not control the size of the test at the nominal level) as in appendix Aa or figure 8 in appendix Ae, and for this reason, we will henceforth compare the USP test with the permutation versions of the competing tests.

To give an intuitive explanation of why Pearson's test struggles so much in this example, recall that the $\chi^2$ statistic (1.1) can be regarded as an estimator of the $\chi^2$ divergence (1.2). Since, when $\epsilon > 0$, the only departures from independence occur in the four top-left cells of our contingency table, we should hope that the contributions to the test statistic from these cells would be large, to allow us to reject the null hypothesis. But these are also the cells for which the cell probabilities are highest, so it is likely that the denominators $e_{ij}$ in the test statistic will be large for these cells. In that case, the contributions to the overall test statistic from these cells will be reduced relative to the corresponding contributions to the USP test statistic, for instance, which has no such denominator (or equivalently, the denominator is 1). In fact, the denominators in Pearson's statistic mean that it is designed to have good power against alternatives that depart from independence only in *low* probability cells. The irony of this is that such cells will typically have low cell counts, meaning that the usual ($\chi^2$ quantile) version of the test cannot be trusted.

Our second example is designed to be at the other end of the sparse/dense alternative spectrum: we will perturb all cell probabilities away from a uniform distribution. More precisely, for $\epsilon \geq 0$, we set

$$p_{ij}^{(\epsilon)} = \frac{1}{IJ} + (-1)^{i+j}\epsilon, \tag{3.2}$$

for $i = 1, \ldots, 6$ and $j = 1, \ldots, 8$. When $\epsilon = 0$, this is just the uniform distribution across all cells (which satisfies the null hypothesis of independence), while when $\epsilon > 0$, cells $(i, j)$ with $i + j$ even have slightly higher probability, and those with $i + j$ odd have slightly lower probability; see figure 4 for a pictorial representation. In this example, $D = IJ\epsilon^2$, so again the null hypothesis is only

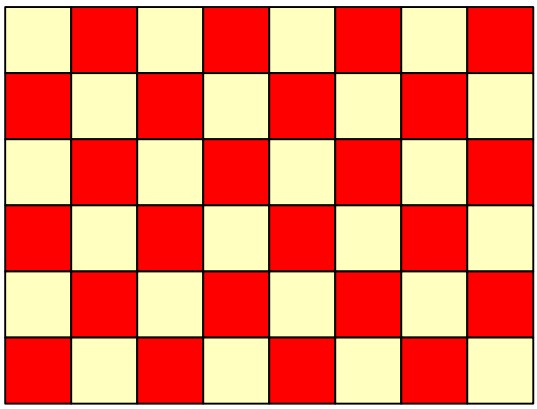

**Figure 4.** Pictorial representation of the cell probabilities in (3.2). (Online version in colour.)

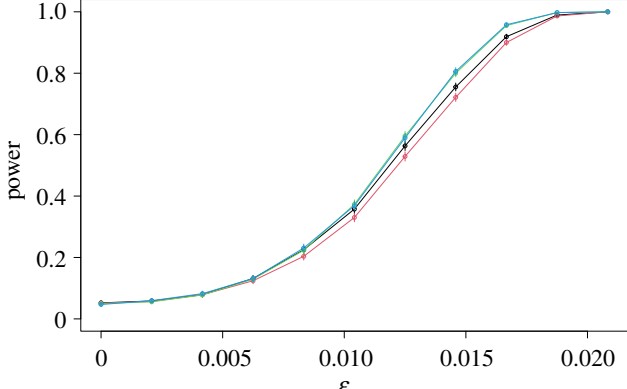

**Figure 5.** Power curves in the dense example, with the USP test in black, Pearson's test in red, the *G*-test in green and Fisher's exact test in cyan. (Online version in colour.)

satisfied when $\epsilon = 0$. Figure 5 plots the power curves, and reveals that all four tests have similar power; in other words, the improved performance of the USP test in the first, sparse example does not come at the expense of worse performance in this dense case. This is not too surprising, because the denominators $e_{ij}$ of Pearson's statistic are nearly constant in this example, so Pearson's statistic is close to a scaled version of the dominant term in the USP test statistic.

Further simulated examples are presented in appendix Ae.

## (c) Real data

Table 3 shows the eye colours of 167 individuals, 85 of whom were female and 82 of whom were male. The *p*-values of the USP test, Pearson's test, the *G*-test and Fisher's exact test were 0.080, 0.169, 0.148 and 0.170, respectively (for the middle two tests, we used the permutation versions of the tests).

To explore this example further, we repeatedly generated further tables of the same size using the empirical cell probabilities from the real data, and computed the proportion of times that the null hypothesis was rejected at the 5% level. Over 1000 repetitions, these proportions were 0.578, 0.491, 0.497 and 0.499 for the USP test, Pearson's test, the *G*-test and Fisher's exact test, respectively, giving further evidence that the USP test is more powerful in this example.

**Table 3.** Contingency table summarizing the eye colours of 85 females and 82 males. Source: www.mathandstatistics.com/learn-stats/probability-and-percentage/using-contingency-tables-for-probability-and-dependence.

|        | black | brown | blue | green | grey |
|--------|-------|-------|------|-------|------|
| female | 20    | 30    | 10   | 15    | 10   |
| male   | 25    | 15    | 12   | 20    | 10   |

**Table 4.** Cell probabilities for our $2 \times 2$ contingency table example.

|         |         |
|---------|---------|
| $p^2$   | $p(1-p)$ |
| $p(1-p)$ | $(1-p)^2$ |

**Table 5.** Cell counts for our $2 \times 2$ contingency table example.

|          |          |
|----------|----------|
| $o_{11}$ | $o_{12}$ |
| $o_{21}$ | $o_{22}$ |

For a second example, we return to the marital status data in table 1. Since the powers for all tests were very high when we resampled as above, we instead repeatedly subsampled 150 observations uniformly at random from the table, again computing the proportion of times that the null hypothesis was rejected at the 5% level. Over 1000 subsamples, the proportions of occasions on which the null hypothesis was rejected at the 5% level were 0.700, 0.583, 0.585 and 0.633 for the USP test, Pearson's test, the $G$-test and Fisher's exact test, respectively, so again the USP test has greatest power over the subsamples.

## 4. Conclusion

$\chi^2$-tests of independence are ubiquitous in scientific studies, but the two most common tests, namely Pearson's test and the $G$-test, can both fail to control the probability of Type I error at the desired level (this can be serious when some cell counts are low), and have poor power. The USP test, by contrast, has guaranteed size control for all sample sizes, can be used without difficulty when there are low or zero cell counts, and has two strong theoretical guarantees related to its power. The first provides a sense in which the USP test is optimal: it is able to detect alternatives for which the measure of dependence $D$ converges to zero at the fastest possible rate as the sample size increases (i.e. no other test could detect alternatives that converge to zero at a faster rate). The second, which is the main new theoretical result of this paper, reveals that the USP test statistic is derived from the unique minimum variance unbiased estimator of $D$. This provides reassurance about the test not just in terms of the rate, but also at the level of constants. These desirable theoretical properties have been shown to translate into excellent performance on both simulated and real data. Specifically, while no test of independence can hope to be most powerful against all departures from independence, we have shown that the USP test is particularly effective when departures from independence occur primarily in high probability cells.

A further extension of our methodology is to the problem of testing homogeneity of the distributions of the different rows of our contingency table. Since the permutations used to generate our $p$-values preserve the marginal row totals, the USP test can be used without modification in this setting, in an analogous way to Pearson's test and the $G$-test.

## (a) An example to show that Pearson's $\chi^2$-test and the $G$-test can have unreliable Type I error

The aim of this subsection is to show that both Pearson's $\chi^2$-test and the $G$-test can have highly unreliable Type I error, even in the simplest setting of a $2 \times 2$ contingency table, and for arbitrarily large sample sizes. Fix a sample size $n$, fix $0 < \lambda < n^{1/2}$, and let $p = \lambda/n^{1/2}$. Consider a $2 \times 2$ contingency table with cell probabilities given in table 4.

It can be checked that this table satisfies the null hypothesis of independence, since

$$P(X = x_1) = p = 1 - P(X = x_2) \quad \text{and} \quad P(Y = y_1) = p = 1 - P(Y = y_2).$$

Now suppose that we draw a random sample of size $n$ from this contingency table, obtaining the cell counts in table 5:

It is convenient to write $\widehat{p}_{i+} = o_{i+}/n$ and $\widehat{p}_{+j} = o_{+j}/n$. Then by some simple but tedious algebra,

$$
\begin{aligned}
\chi^2 &= \frac{(o_{11} - e_{11})^2}{e_{11}} + \frac{(o_{12} - e_{12})^2}{e_{12}} + \frac{(o_{21} - e_{21})^2}{e_{21}} + \frac{(o_{22} - e_{22})^2}{e_{22}} \\
&= \frac{(o_{11} - n\widehat{p}_{1+}\widehat{p}_{+1})^2}{n\widehat{p}_{1+}\widehat{p}_{+1}} + \frac{\{o_{12} - n\widehat{p}_{1+}(1 - \widehat{p}_{+1})\}^2}{n\widehat{p}_{1+}(1 - \widehat{p}_{+1})} + \frac{\{o_{21} - n\widehat{p}_{+1}(1 - \widehat{p}_{1+})\}^2}{n\widehat{p}_{+1}(1 - \widehat{p}_{1+})} \\
&\quad + \frac{\{o_{22} - n(1 - \widehat{p}_{+1})(1 - \widehat{p}_{1+})\}^2}{n(1 - \widehat{p}_{+1})(1 - \widehat{p}_{1+})} \\
&= \frac{(o_{11} - n\widehat{p}_{1+}\widehat{p}_{+1})^2}{n\widehat{p}_{1+}\widehat{p}_{+1}} + \frac{(n\widehat{p}_{1+}\widehat{p}_{+1} - o_{11})^2}{n\widehat{p}_{1+}(1 - \widehat{p}_{+1})} + \frac{(n\widehat{p}_{1+}\widehat{p}_{+1} - o_{11})^2}{n\widehat{p}_{+1}(1 - \widehat{p}_{1+})} + \frac{(n\widehat{p}_{1+}\widehat{p}_{+1} - o_{11})^2}{n(1 - \widehat{p}_{+1})(1 - \widehat{p}_{1+})} \\
&= \frac{(o_{11} - n\widehat{p}_{1+}\widehat{p}_{+1})^2}{n\widehat{p}_{1+}\widehat{p}_{+1}(1 - \widehat{p}_{+1})(1 - \widehat{p}_{1+})} = \frac{(o_{11} - e_{11})^2}{e_{11}(1 - \widehat{p}_{+1})(1 - \widehat{p}_{1+})}. \quad (A\,1)
\end{aligned}
$$

We are now in a position to study the asymptotic distribution of the $\chi^2$ statistic in this model, when $n$ is large and $\lambda$ is fixed. First, note that $o_{11}$, the number of observations in the top-left cell, has a binomial distribution with parameters $n$ and $p^2 = \lambda^2/n$, so its limiting distribution is Poisson with parameter $\lambda^2$, by the law of small numbers (e.g. [14, pp. 2–3]). On the other hand, the other terms in the final expression in (A 1) are converging to constants: $\widehat{p}_{1+}$, the proportion of observations in the first row of the table, is converging to zero in the sense that $P(\widehat{p}_{1+} > t) \to 0$ as $n \to \infty$ for every $t > 0$, and likewise for $\widehat{p}_{+1}$, the proportion of observations in the first column. Finally, we turn to $e_{11}$, and note that we can write $e_{11} = (n^{1/2}\widehat{p}_{1+})(n^{1/2}\widehat{p}_{+1})$. Now, $n^{1/2}\widehat{p}_{1+}$ has the same distribution as $W/n^{1/2}$, where $W$ has a binomial random variable with parameters $n$ and $p = \lambda/n^{1/2}$. Thus $n^{1/2}\widehat{p}_{1+}$ has expectation $\lambda$ and variance $(\lambda/n^{1/2})(1 - (\lambda/n^{1/2}))$, which converges to zero as $n \to \infty$. Since $n^{1/2}\widehat{p}_{+1}$ has the same distribution as $n^{1/2}\widehat{p}_{1+}$, we deduce that $e_{11} = \lambda^2 + E_n$, where $E_n$ converges to zero in the same sense as $\widehat{p}_{1+}$. These calculations allow us to conclude that the asymptotic distribution of the $\chi^2$ statistic in this example is that of

$$\frac{(Z - \lambda^2)^2}{\lambda^2},$$

where $Z$ has a Poisson distribution with parameter $\lambda^2$. We can immediately see from this that, even in the limit as $n \to \infty$, Pearson's test will not have the desired Type I error probability, because this distribution differs from the $\chi^2$ distribution with 1 d.f., which is what would be expected according to the traditional asymptotic theory where the cell probabilities do not change with the sample size. As another way of comparing the actual asymptotic Type I error probability

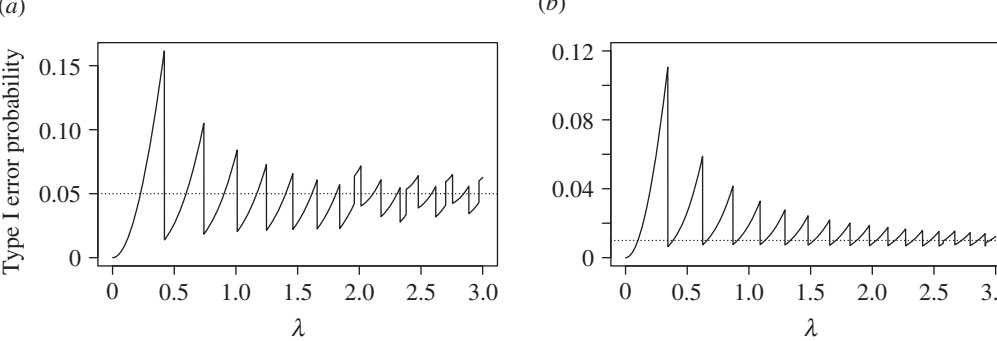

**Figure 6.** Plots of the asymptotic Type I error of Pearson's test for the $2 \times 2$ table with cell probabilities given in table 4 when $\alpha = 0.05$ (*a*) and $\alpha = 0.01$ (*b*).

with the desired level, see figure 6. Here, we plot the asymptotic Type I error probability

$$P\left(\frac{(Z - \lambda^2)^2}{\lambda^2} > c_\alpha\right),$$

as a function of $\lambda$, where $c_\alpha$ is the $(1 - \alpha)$th quantile of the $\chi_1^2$ distribution. For an ideal test of exact size $\alpha$, this should produce a constant flat line at level $\alpha$, but in fact we see that the Type I error probability oscillates quite wildly, due to the discreteness of the Poisson distribution. For a test at a desired 1% significance level, we may end up with a test whose Type I error probability is 10 times larger!

These issues are not resolved by working with the *G*-test instead. Indeed, similar but more involved calculations, given in appendix Ad, reveal that in this example, the asymptotic distribution of the *G*-test statistic is that of

$$2Z \log\left(\frac{Z}{\lambda^2}\right) - 2(Z - \lambda^2),$$

where $Z$ has a Poisson distribution with parameter $\lambda^2$. Since this asymptotic distribution is not a $\chi^2$ distribution with 1 d.f., we again see that the asymptotic size of the *G*-test will not be correct in general. The corresponding asymptotic size plots, which are presented in figure 7, reveal similarly wild behaviour as for Pearson's test. The biggest jumps in the Type I error probabilities in figure 7 occur when $\lambda = \sqrt{c_\alpha}/2$, because when $\lambda$ exceeds this level, we will reject the null hypothesis on observing $Z = 0$, whereas for smaller $\lambda$ we will not. A similar transition occurs when $\lambda = \sqrt{c_\alpha}$ for Pearson's test in figure 6, though this is barely detectable when $\alpha = 0.01$, in which case $\sqrt{c_\alpha}$ is approximately 2.58.

We conclude from this example that the sizes of both Pearson's test and the *G*-test can be extremely unreliable, even when the overall sample size in the contingency table is very large. Moreover, these problems can be even further exacerbated when we move beyond $2 \times 2$ contingency tables, with asymptotic Type I error probabilities that deviate even further from their desired levels.

To explain what is going on in this example in a more general but abstract way, let $\mathcal{P}$ denote the set of all possible distributions on $2 \times 2$ contingency tables that satisfy the null hypothesis of independence. By, e.g. Fienberg & Gilbert [15], all such distributions have cell probabilities of the form given in table 6 for some $0 \le s \le 1$ and $0 \le t \le 1$.

In our example, we simplified this general case by taking $s = t = p$. The justification for using $c_\alpha$ as the critical value for Pearson's $\chi^2$-test comes from the fact that for each $P$ in the set $\mathcal{P}$, we have that

$$P_P(\chi^2 > c_\alpha) \to \alpha,$$

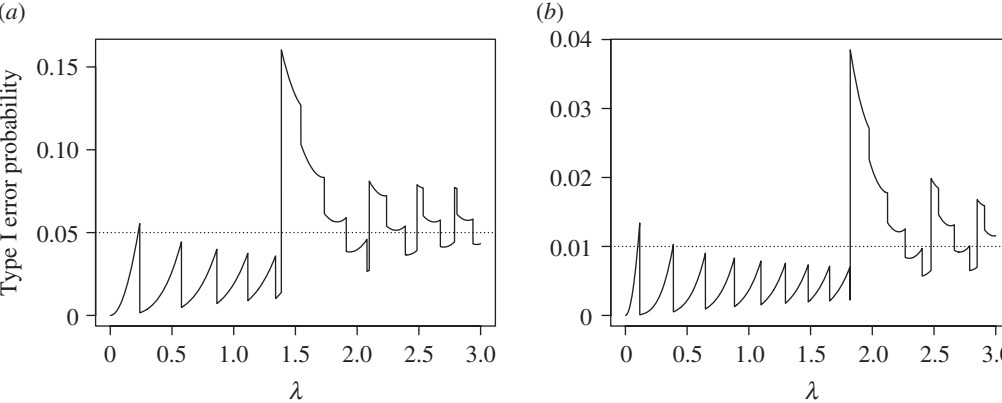

**Figure 7.** Plots of the asymptotic Type I error of the *G*-test for the $2 \times 2$ table with cell probabilities given in table 4 when $\alpha = 0.05$ (*a*) and $\alpha = 0.01$ (*b*).

**Table 6.** Cell probabilities for a general $2 \times 2$ contingency table satisfying the null hypothesis of independence.

| $st$ | $s(1-t)$ |
|---|---|
| $s(1-t)$ | $(1-s)(1-t)$ |

as $n \to \infty$. Here, the notation $P_P$ indicates that the probability is computed under the distribution $P$. On the other hand, the crucial point about our example is that the speed at which this probability converges to $\alpha$ may depend on the particular choice of $P$ that we make; more formally, this convergence is *not* uniform over the class $\mathcal{P}$

$$\sup_{P \in \mathcal{P}} |P_P(\chi^2 > c_\alpha) - \alpha| \not\to 0,$$

as $n \to \infty$. It is this fact that allows us to find, for each $n$, a distribution $P_n$ in $\mathcal{P}$ for which the Type I error probability $P_{P_n}(\chi^2 > c_\alpha)$ is not approaching $\alpha$ as $n$ increases.

## (b) An unbiased estimator of *D*

When $n \geq 4$, the unbiased estimator of $D$ obtained from the fourth-order *U*-statistic of Berrett *et al.* [10] is

$$\widehat{D} = \frac{1}{n(n-3)} \sum_{i=1}^{I} \sum_{j=1}^{J} (o_{ij} - e_{ij})^2 - \frac{4}{n(n-2)(n-3)} \sum_{i=1}^{I} \sum_{j=1}^{J} o_{ij} e_{ij}$$

$$+ \frac{\sum_{i=1}^{I} o_{i+}^2 + \sum_{j=1}^{J} o_{+j}^2}{n(n-1)(n-3)} + \frac{(3n-2)(\sum_{i=1}^{I} o_{i+}^2)(\sum_{j=1}^{J} o_{+j}^2)}{n^3(n-1)(n-2)(n-3)} - \frac{n}{(n-1)(n-3)}. \quad \text{(A 2)}$$

## (c) Proof of theorem 2.1

Since we know that $\widehat{D}$ is an unbiased estimator of $D$, it remains to show that $\widehat{D}$ has minimal variance among all unbiased estimators of $D$ and that no other unbiased estimator of $D$ can match this minimal variance. We may again assume without loss of generality that $X$ takes values in $\{1, \ldots, I\}$ and $Y$ takes values in $\{1, \ldots, J\}$. Our basic strategy is to apply the Lehmann–Scheffé theorem [16,17], which can be regarded as an extension of the Rao–Blackwell theorem [18,19]. The Lehmann–Scheffé theorem relies on the notions of a *sufficient* statistic and a *complete* statistic. Intuitively, a statistic $S$ is sufficient for $D$ if it encapsulates all of the information in the data that

is relevant for making inference about $D$. More formally, $S$ is sufficient in our contingency table setting if the conditional distribution of $((X_1, Y_1), \ldots, (X_n, Y_n))$ given $S$ does not depend on $D$. We claim that the matrix $(o_{ij})$ of observed counts is sufficient for $D$, and this follows because the conditional distribution of interest is given by

$$P(X_1 = x_1, Y_1 = y_1, \ldots, X_n = x_n, Y_n = y_n \mid (o_{ij})) = \frac{\prod_{i=1}^{I} \prod_{j=1}^{J} o_{ij}!}{n!},$$

whenever $\sum_{k=1}^{n} 1_{\{x_k=i, y_k=j\}} = o_{ij}$ for all $i, j$. In other words, once the cell counts are fixed, every ordering of the way in which those cells counts could have arisen is equally likely. This probability does not depend on $D$, so the matrix of observed counts is sufficient for $D$.

Informally, we say $S$ is a complete statistic if there are no non-trivial unbiased estimators of zero that are functions of $S$. This means that whenever $Eg(S) = 0$, we must have $g(S) = 0$. The main part of our proof is devoted to proving that the matrix $(o_{ij})$ is complete. To this end, let $d = IJ$, and let $\Delta$ denote the simplex of all $d$-dimensional probability vectors, so that

$$\Delta = \left\{ (p_1, \ldots, p_d) : p_\ell \geq 0 \text{ for all } \ell \text{ and } \sum_{\ell=1}^{d} p_\ell = 1 \right\}.$$

Note that we are now thinking of stacking the columns of our matrix as a $d$-dimensional vector. We let $N$ denote the set of all possible $d$-dimensional vectors of observed counts with a total sample size of $n$, so that

$$N = \left\{ (n_1, \ldots, n_d) : n_\ell \text{ is a non-negative integer for all } \ell, \text{ and } \sum_{\ell=1}^{d} n_\ell = n \right\}.$$

In the multinomial sampling model for our data, the probability of seeing observed counts $(n_1, \ldots, n_d)$ is

$$f(n_1, \ldots, n_d) = n! \prod_{\ell=1}^{d} \frac{p_\ell^{n_\ell}}{n_\ell!}.$$

Suppose without loss of generality that $p_d > 0$ (if it were zero then we could simply choose a different index). In order to study completeness, we should consider a function $g$ with

$$0 = \sum_{(n_1, \ldots, n_d) \in N} g(n_1, \ldots, n_d) f(n_1, \ldots, n_d)$$

$$= n! \sum_{(n_1, \ldots, n_d) \in N} g(n_1, \ldots, n_d) \frac{(1 - \sum_{\ell=1}^{d-1} p_\ell)^{n_d}}{n_d!} \prod_{\ell=1}^{d-1} \frac{p_\ell^{n_\ell}}{n_\ell!}$$

$$= \frac{n!}{n_d!} \left(1 - \sum_{\ell=1}^{d-1} p_\ell\right)^n \sum_{(n_1, \ldots, n_d) \in N} g(n_1, \ldots, n_d) \prod_{\ell=1}^{d-1} \frac{1}{n_\ell!} \left(\frac{p_\ell}{1 - \sum_{\ell=1}^{d-1} p_\ell}\right)^{n_\ell}. \tag{A 3}$$

Here, in moving from the first line to the second, we have used the fact that $p_d = 1 - \sum_{\ell=1}^{d-1} p_\ell$, and in the final step, we exploited the fact that $n_d = n - \sum_{\ell=1}^{d-1} n_\ell$. Now let

$$z_\ell = \frac{p_\ell}{1 - \sum_{\ell=1}^{d-1} p_\ell},$$

for $\ell = 1, \ldots, d - 1$, and note that each $z_\ell$ can take any non-negative real value if we choose the probabilities $p_1, \ldots, p_{d-1}$ appropriately. Then from (A 3), we deduce that

$$\sum_{(n_1, \ldots, n_d) \in N} g(n_1, \ldots, n_d) \prod_{\ell=1}^{d-1} \frac{z_\ell}{n_\ell!} = 0.$$

Thus, a polynomial in $z_1, \ldots, z_{d-1}$ is identically zero, so its coefficients must be zero. Hence, $g(n_1, \ldots, n_d) = 0$ for all $(n_1, \ldots, n_d) \in N$, so our matrix of observed counts is complete.

The Lehmann–Scheffé theorem states that an unbiased estimator that is a function of a complete, sufficient statistic is the unique minimum variance unbiased estimator. Our estimator $\widehat{D}$ is an unbiased estimator of $D$ that depends on the data $(X_1, Y_1), \ldots, (X_n, Y_n)$ only through the matrix $(o_{ij})$ of observed counts, which is a complete, sufficient statistic, so $\widehat{D}$ is indeed the unique minimum variance unbiased estimator of $D$.

## (d) Asymptotic distribution of the $G$-test statistic

We return to the $2 \times 2$ contingency table example of appendix Aa. We claim that the asymptotic distribution of the four-dimensional standardized multinomial random vector

$$Y_n = \left(o_{11} - \lambda^2, \frac{o_{12} - n^{1/2}\lambda(1 - \lambda/n^{1/2})}{n^{1/4}}, \frac{o_{21} - n^{1/2}\lambda(1 - \lambda/n^{1/2})}{n^{1/4}}, \frac{o_{22} - n(1 - \lambda/n^{1/2})^2}{n^{1/4}}\right), \quad \text{(A 4)}$$

is that of $(Z_1, Z_2, Z_3, Z_4)$, where $Z_1$ and $(Z_2, Z_3, Z_4)$ are independent, with $Z_1$ having a centred Poisson distribution with parameter $\lambda^2$ and with $(Z_2, Z_3, Z_4)$ having a trivariate normal distribution with mean vector zero and singular covariance matrix

$$\Sigma = \begin{pmatrix} \lambda & 0 & -\lambda \\ 0 & \lambda & -\lambda \\ -\lambda & -\lambda & 2\lambda \end{pmatrix}.$$

To see this, note that the random vector can be written as a sum of independent and identically distributed random variables as

$$Y_n = \sum_{i=1}^{n} \left(W_{i1} - \frac{\lambda^2}{n}, \frac{W_{i2} - (\lambda/n^{1/2})(1 - (\lambda/n^{1/2}))}{n^{1/4}}, \right.$$
$$\left. \frac{W_{i3} - (\lambda/n^{1/2})(1 - (\lambda/n^{1/2}))}{n^{1/4}}, \frac{W_{i4} - (1 - (\lambda/n^{1/2}))^2}{n^{1/4}}\right), \quad \text{(A 5)}$$

where, for instance, $W_{i1} = 1_{\{X_i = x_1, Y_i = y_1\}}$. One way to study the asymptotic distribution of $Y_n$, then, is to compute its limiting moment generating function, which is the moment generating function of each summand in (A 5) raised to the power $n$. It will also be convenient to have notation for terms that will be asymptotically negligible: if $(a_n)$ and $(b_n)$ are sequences, we write $a_n = o(b_n)$ if $a_n/b_n \to 0$ as $n \to \infty$; thus $n^{-5/4} = o(n^{-1})$, for example. Each summand in (A 5) can take four possible values, since $(W_{i1}, W_{i2}, W_{i3}, W_{i4})$ must be one of $(1, 0, 0, 0)$, $(0, 1, 0, 0)$, $(0, 0, 1, 0)$ or $(0, 0, 0, 1)$. Now fixing $u = (u_1, u_2, u_3, u_4)^\top \in \mathbb{R}^4$, we can compute as follows:

$$\mathbb{E}(e^{Y_n^\top u}) = e^{-\lambda^2 u_1} \left\{ \frac{\lambda^2}{n} e^{(1,0,0,0)u} + \frac{\lambda}{n^{1/2}}\left(1 - \frac{\lambda}{n^{1/2}}\right) e^{(0, n^{-1/4}, 0, -n^{-1/4})u} \right.$$

$$+ \frac{\lambda}{n^{1/2}}\left(1 - \frac{\lambda}{n^{1/2}}\right) e^{(0, 0, n^{-1/4}, -n^{-1/4})u}$$

$$\left. + \left(1 - \frac{\lambda}{n^{1/2}}\right)^2 e^{(0, -\lambda/n^{3/4}, -\lambda/n^{3/4}, 2\lambda/n^{3/4})u} + o(n^{-1}) \right\}^n$$

$$= e^{-\lambda^2 u_1} \left\{ \frac{\lambda^2}{n} e^{u_1} + \frac{\lambda}{n^{1/2}}\left(1 + \frac{u_2}{n^{1/4}} + \frac{u_2^2}{2n^{1/2}} - \frac{u_4}{n^{1/4}} + \frac{u_4^2}{2n^{1/2}} - \frac{u_2 u_4}{2n^{1/2}}\right) - \frac{\lambda^2}{n} \right.$$

$$+ \frac{\lambda}{n^{1/2}}\left(1 + \frac{u_3}{n^{1/4}} + \frac{u_3^2}{2n^{1/2}} - \frac{u_4}{n^{1/4}} + \frac{u_4^2}{2n^{1/2}} - \frac{u_3 u_4}{2n^{1/2}}\right) - \frac{\lambda^2}{n}$$

$$\left. + 1 - \frac{2\lambda}{n^{1/2}} + \frac{\lambda^2}{n} - \frac{\lambda u_2}{n^{3/4}} - \frac{\lambda u_3}{n^{3/4}} + \frac{2\lambda u_4}{n^{3/4}} + o(n^{-1}) \right\}^n$$

$$\to \exp\left\{ \lambda^2(e^{u_1} - 1) - \lambda^2 u_1 + \frac{\lambda}{2}(u_2^2 + u_3^2 + 2u_4^2 - u_2 u_4 - u_3 u_4) \right\}.$$

This limiting moment generating function agrees with the moment generating function of $(Z_1, Z_2, Z_3, Z_4)$, and therefore establishes the claimed asymptotic distribution (e.g. [20, p. 390]).

In other words,

$$\begin{pmatrix} o_{11} & o_{12} \\ o_{21} & o_{22} \end{pmatrix} = \begin{pmatrix} \lambda^2 + Z_{n1} & n^{1/2}\lambda + n^{1/4}Z_{n2} - \lambda^2 \\ n^{1/2}\lambda + n^{1/4}Z_{n3} - \lambda^2 & n - 2n^{1/2}\lambda + n^{1/4}Z_{n4} + \lambda^2 \end{pmatrix}, \tag{A 6}$$

where the distribution of $(Z_{n1}, Z_{n2}, Z_{n3}, Z_{n4})$ converges to that of $(Z_1, Z_2, Z_3, Z_4)$ as $n \to \infty$. Note that, since the sum of the entries of this matrix is $n$, we must have that $Z_{n1} + n^{1/4}(Z_{n2} + Z_{n3} + Z_{n4}) = 0$. Similarly to our 'little o' notation for negligible deterministic sequences, we now introduce a notation for negligible random sequences: we write $V_n = o_p(1)$ if $P(|V_n| > t) \to 0$ as $n \to \infty$, for every $t > 0$. We can now calculate further that

$$
\begin{aligned}
e_{11} &= \frac{(o_{11} + o_{12})(o_{11} + o_{21})}{n} \\
&= \frac{(n^{1/2}\lambda + n^{1/4}Z_{n2} + Z_{n1})(n^{1/2}\lambda + n^{1/4}Z_{n3} + Z_{n1})}{n} = \lambda^2 + o_p(1); \\
e_{12} &= \frac{(o_{12} + o_{11})(o_{12} + o_{22})}{n} \\
&= \frac{(n^{1/2}\lambda + n^{1/4}Z_{n2} + Z_{n1})(n - n^{1/2}\lambda + n^{1/4}Z_{n2} + n^{1/4}Z_{n4})}{n} \\
&= n^{1/2}\lambda + n^{1/4}Z_{n2} + Z_{n1} - \lambda^2 + o_p(1)
\end{aligned}
$$

and

$$
\begin{aligned}
e_{22} &= \frac{(o_{22} + o_{12})(o_{22} + o_{21})}{n} \\
&= \frac{(n - n^{1/2}\lambda + n^{1/4}Z_{n2} + n^{1/4}Z_{n4})(n - n^{1/2}\lambda + n^{1/4}Z_{n3} + n^{1/4}Z_{n4})}{n} \\
&= n - 2n^{1/2}\lambda + n^{1/4}Z_{n4} + \lambda^2 - Z_{n1} + o_p(1).
\end{aligned}
$$

Hence

$$
\begin{aligned}
G = {}& 2(\lambda^2 + Z_{n1}) \log\left( \frac{\lambda^2 + Z_{n1}}{\lambda^2 + o_p(1)} \right) \\
& + 2(n^{1/2}\lambda + n^{1/4}Z_{n2} - \lambda^2) \log\left( \frac{n^{1/2}\lambda + n^{1/4}Z_{n2} - \lambda^2}{n^{1/2}\lambda + n^{1/4}Z_{n2} - \lambda^2 + Z_{n1} + o_p(1)} \right) \\
& + 2(n^{1/2}\lambda + n^{1/4}Z_{n3} - \lambda^2) \log\left( \frac{n^{1/2}\lambda + n^{1/4}Z_{n3} - \lambda^2}{n^{1/2}\lambda + n^{1/4}Z_{n3} - \lambda^2 + Z_{n1} + o_p(1)} \right) \\
& + 2(n - 2n^{1/2}\lambda + n^{1/4}Z_{n4} + \lambda^2) \log\left( \frac{n - 2n^{1/2}\lambda + n^{1/4}Z_{n4} + \lambda^2}{n - 2n^{1/2}\lambda + n^{1/4}Z_{n4} + \lambda^2 - Z_{n1} + o_p(1)} \right).
\end{aligned}
$$

By a Taylor expansion of the logarithms in the second, third and fourth terms, we conclude that the asymptotic distribution of $G$ is that of

$$2(\lambda^2 + Z_1) \log\left( 1 + \frac{Z_1}{\lambda^2} \right) - 2Z_1,$$

as claimed in appendix Aa.

## (e) Additional simulation results

Here, we present further numerical comparisons between the USP test, Pearson's test, the $G$-test and Fisher's exact test. Figure 8 shows power functions for the first (sparse alternative) example in §2b, but with $n = 400$ instead of $n = 100$. The figure is qualitatively similar in most respects to figure 3, and reveals that the improved performance of the USP test is not diminished by increasing the sample size. One slight difference is that we can see that the version of Pearson's test with the $\chi^2$ quantile is anti-conservative (fails to control the size at the nominal level) for this sample size.

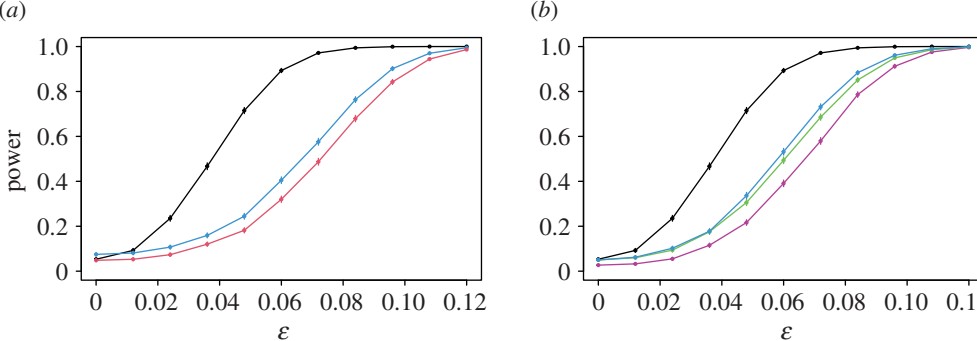

**Figure 8.** Power curves of the USP test in the sparse example with $n = 400$, compared with Pearson's test (*a*) and both the *G*-test and Fisher's exact test (*b*). In each case, the power of the USP test is given in black. The power functions of the $\chi^2$ quantile versions of the first two comparators are shown in blue (*a*) and purple (*b*), while those of the permutation versions of these tests are given in red (*a*) and green (*b*). The power curve of Fisher's exact test is shown in cyan on the right. (Online version in colour.)

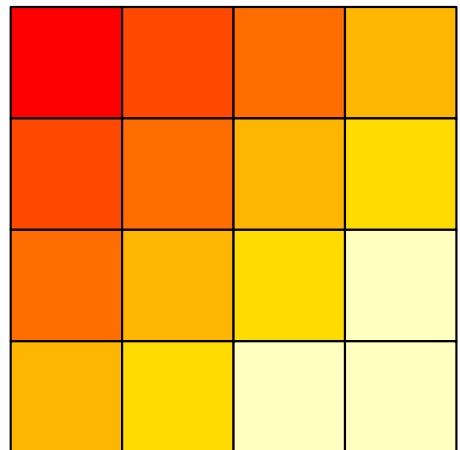

**Figure 9.** Pictorial representation of the cell probabilities in (A 7). (Online version in colour.)

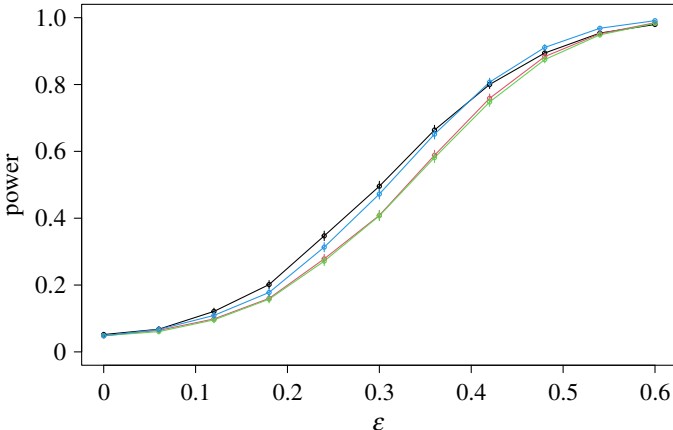

**Figure 10.** Power curves of the USP test (black), Pearson's test (red), the *G*-test (green) and Fisher's exact test (cyan) for the multiplicative example with $n = 100$. (Online version in colour.)

A feature of both our sparse and dense examples is that the perturbations from the null distribution are *additive*. An alternative mechanism for departing from the null distribution that is also of interest is where the perturbations are *multiplicative*. For example, for $I = J = 4$ and $\epsilon \geq 0$, consider the cell probabilities

$$p_{ij}^{(\epsilon)} = \frac{1 + (-1)^{i+j}\epsilon}{C_\epsilon \cdot 2^{i+j}}, \tag{A 7}$$

where $C_\epsilon = \sum_{i=1}^{I} \sum_{j=1}^{J} (1 + (-1)^{i+j}\epsilon/2^{i+j})$ is a normalization constant; see figure 9. Figure 10 shows the power curves of our four permutation tests with $n = 100$. Despite the fact that the perturbations here are dense, we see that the USP test is best able to detect the violations of independence for small and moderate values of $\epsilon$, while Fisher's test slightly outperforms it for larger $\epsilon$.

Data accessibility. This article has no additional data.

Authors' contributions. T.B.B. conceived of the general framework and helped draft the manuscript. R.J.S. helped formulate the general framework and drafted the manuscript. Both authors gave final approval for publication and agree to be held accountable for the work performed therein.

Competing interests. We declare we have no competing interests.

Funding. The research of R.J.S. was supported by EPSRC grant nos EP/P031447/1 and EP/N031938/1, as well as ERC grant no. 101019498. The authors are grateful for helpful feedback from Sergio Bacallado, Rajen Shah and Qingyuan Zhao.

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
