## [Peer Review File · Proceedings. Mathematical, Physical, and Engineering Sciences]

Review History

RSPA-2021-0549.R0 (Original submission)

Review form: Referee 1

Is the manuscript an original and important contribution to its field?

Excellent

Is the paper of sufficient general interest?

Good

Is the overall quality of the paper suitable?

Excellent

Can the paper be shortened without overall detriment to the main message?

Yes

Do you think some of the material would be more appropriate as an electronic appendix?

No

Do you have any ethical concerns with this paper?

No

Recommendation?

Accept with minor revision (please list in comments)

Comments to the Author(s)

All the comments are in the attached file below. (See Appendix A)

Review form: Referee 2

Is the manuscript an original and important contribution to its field?

Good

Is the paper of sufficient general interest?

Good

Is the overall quality of the paper suitable?

Good

Can the paper be shortened without overall detriment to the main message?

Yes

Do you think some of the material would be more appropriate as an electronic appendix?

No

Do you have any ethical concerns with this paper?

No

Recommendation?

Accept with minor revision (please list in comments)

Comments to the Author(s)

This paper is well written and a useful complement to the more technical paper "Optimal rates for independence testing via U-statistic permutation tests" which is due to appear. The focus of this paper is to consider a U-statistic permutation test for independence that does not face the same limitations, e.g. problems when cells with small counts are evident, as methods such as Pearson's chi-squared test. I have a few main comments/queries below.

Other permutation tests are available and perhaps this needs to be mentioned. For example, there is the `perm.ind.test` in the R package `wPerm`. Maybe this should be compared to the USP as well, or made clear why it shouldn't.

For the example in Section 3.3 (eye color, gender) I could not get a p-value as low as 0.0495. In fact over 100 and $B = 999$, estimations the smallest p-value I obtained was closer to 0.07. For $B = 9999$ so as to get a more stable p-value the smallest I obtained was nearly 0.08. Is this a mistake? What choice of B was used? E.g. I was able to obtain a p-value this low when using $B = 99$ but I don't anyone who choose a B this low. Also, I have another related comment below on this.

Since it is a permutation test, the p-value will be different each time it is carried out even on the same data set. For smaller choices B , the p-value is quite variable. While I am not in favor of simply a hard and fast reject/do not reject decisions for a set level of significance, the reality is that many will use this in that way. For the example in Table 3 and $B = 999$, I obtain p-values of anywhere between 0.07 and 0.11. For $B = 9999$ between 0.08 and 0.09. Obviously this is trait of permutation tests in general, but since the test could be misused by some, it would be good for there to be some discussion or guidance on this.

Page 14, Table 3.3 example again. To compare power, 84 observations were repeatedly sub-sampled. Why 84? And wouldn't an option have been to sample 167 where observations are sampled according to the cell probabilities set to the observed proportions.

Decision letter (RSPA-2021-0549.R0)

02-Nov-2021

Dear Professor Samworth,

On behalf of the Editor, I am pleased to inform you that your Manuscript RSPA-2021-0549 entitled "USP: an independence test that improves on Pearson's chi-squared and the G-test" has been accepted for publication subject to minor revisions in Proceedings A. Please find the referees' comments below.

The reviewer(s) have recommended publication, but also suggest some minor revisions to your manuscript. Therefore, I invite you to respond to the reviewer(s)' comments and revise your manuscript. Please note that we have a strict upper limit of 28 pages for each paper. Please endeavour to incorporate any revisions while keeping the paper within journal limits. Please note that page charges are made on all papers longer than 20 pages. If you cannot pay these charges you must reduce your paper to 20 pages before submitting your revision. Your paper has been ESTIMATED to be 27 pages. We cannot proceed with typesetting your paper without your agreement to meet page charges in full should the paper exceed 20 pages when typeset. If you have any questions, please do get in touch.

It is a condition of publication that you submit the revised version of your manuscript within 7 days. If you do not think you will be able to meet this date please let me know in advance of the due date.

To revise your manuscript, log into <https://mc.manuscriptcentral.com/prsa> and enter your Author Centre, where you will find your manuscript title listed under "Manuscripts with Decisions." Under "Actions," click on "Create a Revision." Your manuscript number has been appended to denote a revision.

You will be unable to make your revisions on the originally submitted version of the manuscript. Instead, revise your manuscript and upload a new version through your Author Centre.

When submitting your revised manuscript, you will be able to respond to the comments made by the referee(s) and upload a file "Response to Referees" in Step 1: "View and Respond to Decision Letter". Please provide a point-by-point response to the comments raised by the reviewers and the editor(s). A thorough response to these points will help us to assess your revision quickly. You can also upload a 'tracked changes' version either as part of the 'Response to reviews' or as a 'Main document'.

IMPORTANT: Your original files are available to you when you upload your revised manuscript. Please delete any redundant files before completing the submission process.

When uploading your revised files, please make sure that you include the following as we cannot proceed without these:

- 1) A text file of the manuscript (doc, txt, rtf or tex), including the references, tables (including captions) and figure captions. Please remove any tracked changes from the text before submission. PDF files are not an accepted format for the "Main Document".

2) A separate electronic file of each figure (tif, eps or print-quality pdf preferred). The format should be produced directly from original creation package, or original software format.

3) Electronic Supplementary Material (ESM): all supplementary materials accompanying an accepted article will be treated as in their final form. Note that the Royal Society will not edit or typeset supplementary material and it will be hosted as provided. Please ensure that the supplementary material includes the paper details where possible (authors, article title, journal name). Supplementary files will be published alongside the paper on the journal website and posted on the online figshare repository (<https://figshare.com>). The heading and legend provided for each supplementary file during the submission process will be used to create the figshare page, so please ensure these are accurate and informative so that your files can be found in searches. Files on figshare will be made available approximately one week before the accompanying article so that the supplementary material can be attributed a unique DOI. Alternatively you may upload a zip folder containing all source files for your manuscript as described above with a PDF as your "Main Document". This should be the full paper as it appears when compiled from the individual files supplied in the zip folder.

Article Funder

Please ensure you fill in the Article Funder question on page 2 to ensure the correct data is collected for FundRef (<http://www.crossref.org/fundref/>).

Media summary

Please ensure you include a short non-technical summary (up to 100 words) of the key findings/importance of your paper. This will be used for to promote your work and marketing purposes (e.g. press releases). The summary should be prepared using the following guidelines:

*Write simple English: this is intended for the general public. Please explain any essential technical terms in a short and simple manner.

*Describe (a) the study (b) its key findings and (c) its implications.

*State why this work is newsworthy, be concise and do not overstate (true 'breakthroughs' are a rarity).

*Ensure that you include valid contact details for the lead author (institutional address, email address, telephone number).

Cover images

We welcome submissions of images for possible use on the cover of Proceedings A. Images should be square in dimension and please ensure that you obtain all relevant copyright permissions before submitting the image to us. If you would like to submit an image for consideration please send your image to proceedingsa@royalsociety.org

Open Access

You are invited to opt for open access, our author pays publishing model. Payment of open access fees will enable your article to be made freely available via the Royal Society website as soon as it is ready for publication. For more information about open access please visit <https://royalsociety.org/journals/authors/open-access/>. The open access fee for this journal is £1700/\$2380/€2040 per article. VAT will be charged where applicable. Please note that if the corresponding author is at an institution that is part of a Read and Publishing deal you are required to select this option. See <https://royalsociety.org/journals/librarians/purchasing/read-and-publish/read-publish-agreements/> for further details.

Once again, thank you for submitting your manuscript to Proceedings A and I look forward to receiving your revision. If you have any questions at all, please do not hesitate to get in touch.

Best wishes
Raminder Shergill
proceedingsa@royalsociety.org
Proceedings A

on behalf of
Professor Matjaz Perc
Board Member
Proceedings A

Reviewer(s)' Comments to Author:
Referee: 1
Comments to the Author(s)
All the comments are in the attached file below.

Referee: 2
Comments to the Author(s)

This paper is well written and a useful complement to the more technical paper "Optimal rates for independence testing via U-statistic permutation tests" which is due to appear. The focus of this paper is to consider a U-statistic permutation test for independence that does not face the same limitations, e.g. problems when cells with small counts are evident, as methods such as Pearson's chi-squared test. I have a few main comments/queries below.

Other permutation tests are available and perhaps this needs to be mentioned. For example, there is the `perm.ind.test` in the R package `wPerm`. Maybe this should be compared to the USP as well, or made clear why it shouldn't.

For the example in Section 3.3 (eye color, gender) I could not get a p-value as low as 0.0495. In fact over 100 and $B = 999$, estimations the smallest p-value I obtained was closer to 0.07. For $B = 9999$ so as to get a more stable p-value the smallest I obtained was nearly 0.08. Is this a mistake? What choice of B was used? E.g. I was able to obtain a p-value this low when using $B = 99$ but I don't anyone who choose a B this low. Also, I have another related comment below on this.

Since it is a permutation test, the p-value will be different each time it is carried out even on the same data set. For smaller choices B , the p-value is quite variable. While I am not in favor of simply a hard and fast reject/do not reject decisions for a set level of significance, the reality is that many will use this in that way. For the example in Table 3 and $B = 999$, I obtain p-values of anywhere between 0.07 and 0.11. For $B = 9999$ between 0.08 and 0.09. Obviously this is trait of permutation tests in general, but since the test could be misused by some, it would be good for there to be some discussion or guidance on this.

Page 14, Table 3.3 example again. To compare power, 84 observations were repeatedly sub-sampled. Why 84? And wouldn't an option have been to sample 167 where observations are sampled according to the cell probabilities set to the observed proportions.

Decision letter (RSPA-2021-0549.R1)

10-Nov-2021

Dear Professor Samworth

I am pleased to inform you that your manuscript entitled "USP: an independence test that improves on Pearson's chi-squared and the G-test" has been accepted in its final form for publication in Proceedings A.

Our Production Office will be in contact with you in due course. You can expect to receive a proof of your article soon. Please contact the office to let us know if you are likely to be away from e-mail in the near future. If you do not notify us and comments are not received within 5 days of sending the proof, we may publish the paper as it stands.

As a reminder, you have provided the following 'Data accessibility statement' (if applicable). Please remember to make any data sets live prior to publication, and update any links as needed when you receive a proof to check. It is good practice to also add data sets to your reference list. Statement (if applicable):

Under the terms of our licence to publish you may post the author generated postprint (ie. your accepted version not the final typeset version) of your manuscript at any time and this can be made freely available. Postprints can be deposited on a personal or institutional website, or a recognised server/repository. Please note however, that the reporting of postprints is subject to a media embargo, and that the status the manuscript should be made clear. Upon publication of the definitive version on the publisher's site, full details and a link should be added.

You can cite the article in advance of publication using its DOI. The DOI will take the form: 10.1098/rspa.XXXX.YYYY, where XXXX and YYYY are the last 8 digits of your manuscript number (eg. if your manuscript number is RSPA-2017-1234 the DOI would be 10.1098/rspa.2017.1234).

For tips on promoting your accepted paper see our blog post:
<https://royalsociety.org/blog/2020/07/promoting-your-latest-paper-and-tracking-your-results/>

On behalf of the Editor of Proceedings A, we look forward to your continued contributions to the Journal.

Sincerely,
Raminder Shergill
proceedingsa@royalsociety.org

on behalf of
Professor Matjaz Perc
Board Member
Proceedings A

Appendix A

Revision of Manuscript RSPA-2021-0549

Proceedings of the Royal Society A

It is a very interesting manuscript. They present a U-statistic permutation (USP) test of independence for discrete data displayed in contingency table. They compared the USP test with the Pearson's chi-squared and the G-test and showed that the USP test controls the size of the test at the nominal level for all samples sizes and that it has very good power properties.

1. The Pearson χ^2 -statistic and the G-statistic can be used for the hypothesis of homogeneity and independence. Can this test statistic be used for the hypothesis of homogeneity as well?
2. What is the difference (advantage/desadvantage) of the USP test performance using resampling techniques such as bootstrap and a fixed number of permutations?
3. p.3 line 8 - It may be useful to mention that the G-statistic is approximately equal to the Pearson's χ^2 -statistic by Taylor expansion (using $\ln(1+x) \approx x - \frac{1}{2}x^2 + O(x^2)$).
4. p.4 lines 7-23 - It may be useful to mention some comments of Agresti, 2007, Section 2.4.7 (p.40) and Section 2.6 about exact tests. Also, update the reference: Agresti, A. (2007) - "An Introduction to Categorical Data Analysis". 2nd edition. John Wiley & Sons. Hoboken, New Jersey.
5. p.7 - Write \hat{D} as a U-statistic of degree 4, i.e.,

$$\hat{D} = \frac{1}{4! \binom{n}{4}} \sum_{(i_1, i_2, i_3, i_4)} h[(X_{i_1}, Y_{i_1}), (X_{i_2}, Y_{i_2}), (X_{i_3}, Y_{i_3}), (X_{i_4}, Y_{i_4})]$$

Then, refer to Section 5.2 (equation (7)) and the explanation of why \hat{D} can be simplified to \hat{U} (page 8, lines 32-46).

6. In Section 3 - Numerical Results, it would be nice to provide the p-value for the Fisher's exact test to compare with the p-values of the USP, Pearson's chi-square and G-test.
7. p. 10 line 25 - It should be "A pictorial representation..."
8. For small samples sizes and/or sparse examples, it would be nice to include the Fisher's exact test in the power comparison.